# Glycodelin as a Serum and Tissue Biomarker for Metastatic and Advanced NSCLC

**DOI:** 10.3390/cancers10120486

**Published:** 2018-12-04

**Authors:** Marc A. Schneider, Thomas Muley, Rebecca Weber, Sabine Wessels, Michael Thomas, Felix J. F. Herth, Nicolas C. Kahn, Ralf Eberhardt, Hauke Winter, Gudula Heussel, Arne Warth, Christel Herold-Mende, Michael Meister

**Affiliations:** 1Translational Research Unit, Thoraxklinik at University Hospital Heidelberg, D-69126 Heidelberg, Germany; thomas.muley@med.uni-heidelberg.de (T.M.); rebecca.weber@med.uni-heidelberg.de (R.W.); michael.meister@med.uni-heidelberg.de (M.M.); 2Translational Lung Research Center Heidelberg (TLRC), Member of the German Center for Lung Research (DZL), D-69120 Heidelberg, Germany; 3Department of Thoracic Oncology, Thoraxklinik at University Hospital Heidelberg, D-69126 Heidelberg, Germany; sabine.wessels@med.uni-heidelberg.de (S.W.); michael.thomas@med.uni-heidelberg.de (M.T.); 4Department of Pneumology and Critical Care Medicine, Thoraxklinik at University Hospital Heidelberg, D-69126 Heidelberg, Germany; felix.herth@med.uni-heidelberg.de (F.J.F.H.); nicolas.kahn@med.uni-heidelberg.de (N.C.K.); ralf.eberhardt@med.uni-heidelberg.de (R.E.); 5Department of Surgery, Thoraxklinik at University Hospital Heidelberg, D-69126 Heidelberg, Germany; hauke.winter@med.uni-heidelberg.de; 6Department of Diagnostic and Interventional Radiology with Nuclear Medicine, Thoraxklinik at University Hospital Heidelberg, D-69126 Heidelberg, Germany; gudula.heussel@med.uni-heidelberg.de; 7Department of Diagnostic and Interventional Radiology, University Hospital, D-69120 Heidelberg, Germany; 8Institute of Pathology, University Hospital Heidelberg, D-69120 Heidelberg, Germany; warth@patho-uegp.de; 9Division of Experimental Neurosurgery, Heidelberg University Hospital, D-69120 Heidelberg, Germany; Christel.herold-mende@med.uni-heidelberg.de; 10German Cancer Consortium (DKTK), D-69126 Heidelberg, Germany

**Keywords:** *PAEP*, glycodelin, NSCLC, early detection, follow-up biomarker

## Abstract

A major part of non-small cell lung cancer (NSCLC) patients treated with mono- or multimodal concept develop therapy resistance. Despite the abundance of biomarkers investigated in the past, there is still a need for valid NSCLC biomarkers. Glycodelin, an immunosuppressive endometrial protein, has been shown to be also expressed in NSCLC. Here, we investigated its potential as a biomarker in metastatic and advanced stage NSCLC. Glycodelin gene and protein expression were measured in 28 therapy-naïve resected tumors as well as in corresponding brain (*n* = 16) and adrenal gland (*n* = 12) metastasis by qPCR and IHC. Moreover, we correlated glycodelin gene expression of cryoconserved therapy-naïve biopsies (*n* = 55) of advanced stage patients with glycodelin serum concentrations and patient survival. Using follow-up samples of the patients, we monitored glycodelin serum concentrations during therapy. Glycodelin expression correlated between primary tumor and distant metastases within the same patients. The gene expression of glycodelin in therapy-naïve biopsies also correlated with the serum concentrations of the patients (r = 0.60). Patients with elevated serum concentrations showed a tendency in lower overall survival (*p* = 0.088) and measuring of glycodelin indicated a progression of the disease earlier compared to clinical diagnostic. Taken together, we demonstrate that glycodelin is a promising prognostic and follow-up biomarker for metastatic and advanced NSCLC.

## 1. Introduction

Lung cancer is the most common cause for cancer related death worldwide [1]. Since most cases are detected in advanced stages, the survival rates are poor [2]. In recent years, targeted therapies as well as therapies with immune checkpoint inhibitors have supplemented multimodal palliative treatment. These new therapies have led to an improvement of disease free and overall survival of patients with metastasized tumors [3]. Nevertheless, most tumors develop resistances during therapy. An early detection of progression is of great importance for further therapy options when therapy resistance occurred. There are only few biomarkers known which have been demonstrated to predict information about a response or progression during therapy in NSCLC. The carcinoembryonic antigen-related cell adhesion molecule 5 (CEA) and cytokeratin fragment antigen 21-1 (CYFRA 21-1) for example have been well characterized as NSCLC biomarkers and investigated for more than 20 years [4,5]. Detection of free circulating tumor DNA (cfDNA) has also been shown to be an option in advanced stage NSCLC patients with reduced general conditions [6,7]. Nevertheless, the detection rates of DNA mutations in liquid biopsies in general are much lower compared to biopsies. Therefore, there is still a great necessity for NSCLC serum biomarkers.

Glycodelin (gene name progesterone associated endometrial protein (*PAEP*)) has been well described during the menstrual cycle and pregnancy [8,9]. It has been shown to be expressed by the inner layers of the endometrium with high local concentrations and to act as an immunosuppressive protein [9,10]. Glycodelin expression during pregnancy has also been suggested to regulate the invasion deepness of the trophoblast and to be involved in successful implantation [11]. Beside its function during pregnancy and implantation, glycodelin has been shown to be overexpressed in hormone-related cancers, such as ovarian cancer [12] and breast cancer [13]. In part, contrary data has been published for endometrial cancer and breast cancer concerning prognostic influence and molecular functions of glycodelin [14,15]. In the last decade, increased glycodelin expression levels have also been described for hormone-independent malignancies as the malignant melanoma [16], non-small cell lung cancer [17] and the malignant pleural mesothelioma [18]. In 2015, we investigated the expression and function of glycodelin in a large surgical NSCLC cohort [17]. We observed that the gene was overexpressed in approximately 80% of all tumors. While *PAEP* was not expressed in normal lung tissue in half of the patients we examined, it was expressed up to 10,000 fold higher in tumor tissue. Moreover, we were able to detect glycodelin in the serum of approximately 40% of patients with NSCLC. Therefore, we suggested glycodelin as a diagnostic and prognostic biomarker for NSCLC.

The majority of patients suffering from NSCLC are diagnosed at advanced disease stages with occurring metastases. Therefore, the current study addressed the potential of glycodelin as a prognostic and follow-up biomarker in metastatic disease and inoperable stage IIIB/IV patients.

## 2. Results

### 2.1. Glycodelin Expression in Primary Lung Tumor and Adrenal Gland Metastasis

We first investigated the *PAEP* (glycodelin gene) and glycodelin (protein) expression in patients who underwent surgery for the excision of the primary tumor and adrenal gland metastases (Table 1). In general, we observed that *PAEP* was expressed on similar levels comparing the primary tumor and the metastasis (Figure 1A). However, the expression level of *PAEP* varied greatly in 12 patients. In the lung tumor of patient 1, it was approximately 4000 fold higher than in patient 12 (difference of 12 qPCR cycles). In line with that, the *PAEP* expression in the adrenal gland metastases was also very heterogeneous (Figure 1A,B). The *PAEP* gene expression between primary tumor and metastases was not significantly different (Figure 1B).

The correlation analyses demonstrated that the relative gene expression level highly correlated (*r* = 0.85 and *p* = 0.0005) between the primary tumor and the adrenal gland metastases (Figure 1C). Furthermore, we observed a good correlation between the expression of the gene and the protein (Figure 1D). Higher gene expression in principle resulted in a stronger staining in IHC. Thereby, glycodelin was often differentially expressed by the single tumor cells and heterogeneously distributed in the tissue (as shown for patients 4, 5 and 8, Figure 1D).

### 2.2. Glycodelin Expression in Primary Lung Tumor and Brain Metastasis

The observations concerning *PAEP* and glycodelin expression in brain metastases were similar to the adrenal gland metastasis. However, the gene expression displayed a higher variation in brain metastases (Figure 2A). While some metastases samples showed a higher *PAEP* expression (patient 5 and 13, Figure 2A) compared to lung tumors, there were others with much lower gene expression (patient 1, 7 and 8).

Generally, the *PAEP* expression level did not significantly differ in lung tumors and brain metastasis (Figure 2B). The relative RNA expression level in brain metastases was higher than in lung tumors (Figure 2B) and also than in adrenal gland metastasis (ΔCt brain = 5.52 compared to ΔCt adrenal gland = 7.89, Figure 2B and Figure 1B). However, the IHC staining of glycodelin in brain metastases was weaker than in adrenal gland metastasis (Figure 2D). The correlation between *PAEP* expression in primary tumors and brain metastasis was lower compared to the group of patients with adrenal gland metastases, but still significant (*r* = 0.63, *p* = 0.0094, Figure 2C).

### 2.3. Glycodelin Expression in Cryo-Conserved Biopsies of Advanced Stage NSCLC Patients Correlated with Measured Glycodelin Serum Concentrations

Beside the metastasized cohort, we were also interested in glycodelin expression in advanced stage tumors (Table 1) and its potential as a prognostic follow-up biomarker for these patients. Within a cohort assembly of the German Center for Lung Research (DZL), we analyzed the *PAEP* expression of 55 tumor and 17 non-neoplastic lung biopsies. *PAEP* expression was significantly upregulated in the tumor samples (*p* = 0.0029) and showed a wide range of expression (approx. 15 qPCR cycles) compared to non-neoplastic tissues (Figure 3A). 

Here, the expression of *PAEP* was significantly higher in adenocarcinoma (ADC) compared to squamous cell carcinoma (SQCC) (Figure 3B). Since glycodelin is a secretory protein, we additionally investigated the protein levels in the sera of the same patients. In line with observations from an earlier study [17], the tissue gene expression of glycodelin significantly correlated with the measured serum levels (Figure 3C). Furthermore, we observed significant higher serum concentrations (*p* = 0.046) of patients with ADC compared to SQCC (Figure 3D). These data implicate that glycodelin is a suitable tissue and serum biomarker especially for ADC patients.

### 2.4. Glycodelin Serum Concentration Is a Prognostic Factor for the Survival of NSCLC Patients

To investigate its potential as a prognostic serum marker for advanced stages, we analyzed the pretherapeutic glycodelin serum concentration of patients with stage IIIB and IV. The serum was collected at the time of biopsy. The overall survival tended to be lower (*p* = 0.088) at glycodelin serum concentrations >5 ng/mL (Figure 4A). Interestingly, the lower survival was especially seen in female but not in male patients (Figure 4B,C, *p* = 0.071 and *p* = 0.299).

In regard to histology, ADC patients had a significant worse prognosis when higher glycodelin (>5 ng/mL) concentrations were measured (Figure 4D, *p* = 0.036). For patients with SQCC, it seemed to be similar (Figure 4E, *p* = 0.035) but the group with high glycodelin serum levels contained only two patients.

Therefore, these data must be judged with caution. An influence of the age of female patients on glycodelin serum concentrations because of their possibly present menstruation cycle was excluded using correlation analysis (*r* = 0.035, data not shown). An association between driver mutations in ADC (EGFR/ALK/KRAS neg. (*n* = 9) vs. EGFR/ALK/KRAS positive (*n* = 5)) and glycodelin levels was not observed (data not shown). The patient number with other NSCLC histologies (NOS, mixed types, large cell carcinoma, *n* = 9) was too low and not included in statistical analyses. Our data highly support the hypothesis that glycodelin is a feasible prognostic biomarker for NSCLC patients.

### 2.5. Glycodelin as a Follow-Up Biomarker for Monitoring of Advanced Stage NSCLC Patients

As a part of the DZL cohort assembly, serum samples of the patients were routinely collected during their clinical follow-up. Therefore, these samples were well suited to investigate the glycodelin serum concentrations and compare them to clinical outcome. Representative examples of the investigated patients during the course of the disease are given in Figure 5. While a progress was detected in patients 1, 3 and 4, glycodelin serum concentrations dropped down to the baseline value for patient 2. Compared to evaluation of treatment results using CT scan or X-rays, a progression could be detected earlier if glycodelin serum levels were considered. For patient 1, late progression was detected at least 1 month earlier measuring glycodelin than clinical monitoring. Patient 3 already showed increasing glycodelin levels 10 months after diagnosis time while staging defined progression 7 months later. These data strongly suggest the benefit of a frequent measurement of glycodelin during therapy to detect a progression of the disease.

## 3. Discussion

The molecular characterization of NSCLC tumors offers more treatment options for patients resulting in a better overall response rate and prognosis for metastatic and stage IV patients [19,20,21]. Nevertheless, the majority of the patients will develop progressive disease at a certain time point. A closely clinical monitoring using CT is currently used to detect a progression of the disease and new metastases as early as possible. However, in contrast to imaging, the measurement of serum biomarkers is in general a cost effective method with less burden for the patient and therefore more suitable for monitoring disease status. Besides NSCLC markers CEA and CYFRA 21-1 [4,5], many biomarkers are described but have not found their ways into clinical routine diagnostics. Therefore, valuable biomarkers for prognosis and disease follow-up of NSCLC are in demand.

Glycodelin has been well described during implantation of the trophoblast where it has been shown to act as an immunosuppressive protein [9]. We demonstrated for the first time that glycodelin was expressed and secreted by lung tumors and detectable in the sera of patients [17]. Using a surgical cohort, we showed that glycodelin may well be a useful marker for therapy monitoring and detection of disease recurrence.

In this study, we were interested in evaluating the performance of glycodelin in metastatic and advanced stages of the disease. Using surgical tissue, we demonstrated that glycodelin was expressed in adrenal gland and brain metastasis of NSCLC, two major metastasis sites of NSCLC [22]. In addition, the glycodelin RNA expression correlated with the protein expression in primary tumor and metastases. The RNA expression levels of *PAEP* did not significantly differ between primary sites and metastasis. Nevertheless, the gene expression was elevated in the metastases. This might be explained by the immunomodulatory function of glycodelin. Tumor cells undergo immune surveillance during tumor invasion which might be supplemented by the secretion of high local glycodelin concentrations. In line with our observations in an earlier study [17], *PAEP* expression as well as glycodelin serum concentration in patients with ADC were much higher compared to SQCC. SQCC tumors can become very large without resulting in metastatic spread, while ADC have been shown to often spread in early stages of the disease [23,24]. A higher glycodelin expression might therefore increase the immune system suppression to promote metastatic spread of the tumor cells.

The prognosis of NSCLC is always associated with treatment success of the patients. Thus, prognostic biomarkers can support the staging process during therapy. This could be demonstrated for example for the blood markers CEA and CYFRA 21-1 in NSCLC [4,5]. Our data suggested that the survival of NSCLC patients correlated with glycodelin serum levels. However, the cohort size of our study was limited. Our focus was the comparison of glycodelin gene and protein expression in tissue and blood samples from the same patients. Cryoconserved biopsies for research purposes are of high value but sampling is complex. Therefore, future studies with enlarged sample sizes have to validate our findings, especially for the measurement of glycodelin in serum. In addition, a comparison of the prognostic value of glycodelin and other serum markers (e.g., CEA and CYFRA 21-1) might offer the opportunity to combine various prognosis markers. Interestingly, higher serum concentrations (>5 ng/mL) were prognostic only for female but not for male patients, although glycodelin was also highly expressed and detectable in males. The expression and function of glycodelin was well investigated during pregnancy [9]. Glycodelin was also described to be expressed in several hormone related cancers such as ovarian cancer [25] and breast cancer [26]. Therefore, a gender specific effect, influenced by hormone balance, might be conceivable. Since glycodelin has been shown to be expressed during menstruation cycle [8], one must keep in mind that the detection of glycodelin in the context of a malignant disease might be false positive. The median age of the female cohort was 63 whereas the youngest female patient was 52 at time of biomaterial sampling. Most malignancies, such as NSCLC and breast cancer, occur at advanced ages, menstruation cycle should not be an issue for the detection of glycodelin. However, if possible, its status should be stated.

Using follow-up serum samples, we demonstrated that glycodelin measurements can detect progression of the disease earlier and with lower burdens for the patients as compared to imaging. However, a definite conclusion cannot be drawn based on these limited numbers of cases. To validate our findings, this requires further studies. 

During clinical management of the disease, blood samples are taken regularly for routine diagnostic. Therefore, the test of glycodelin is inexpensive and could be easily implemented into clinical setup.

## 4. Materials and Methods

### 4.1. Sampling of Biomaterial

NSCLC cryoconserved tissue was provided by the Lung Biobank Heidelberg, a member of the accredited Tissue Bank of the National Center for Tumor Diseases (NCT) Heidelberg, the BioMaterialBank Heidelberg and the Biobank platform of the German Center for Lung Research (DZL). FFPE-tissue from NSCLC, adrenal gland and brain metastases were provided by the NCT. 

All subjects gave their informed consent for inclusion before they participated in the study. The study was conducted in accordance with the Declaration of Helsinki. The use of biomaterial and data for this study was approved by the local ethics committee of the Medical Faculty Heidelberg (S-048/2012 for biopsies, S-270/2001 and S-005/2003 for surgical material and blood). Tissues are snap-frozen immediately after removal and stored at −80 °C. More detailed information is described elsewhere [27]. Patients’ characteristics are shown in Table 1. 

Serial blood sampling is conducted at baseline and in follow-up. Blood was collected and processed within 1 h. Serum and plasma aliquots were stored at −80 °C until measurements.

### 4.2. Total RNA Isolation and cDNA Synthesis

For RNA isolation from NSCLC and adrenal gland metastasis, a tumor content of ≥50% for surgical tissue and ≥40% for biopsies and brain metastases was the minimum prerequisite. For surgical tissue, 10–15 tumor cryosections (10–15 µM) of each patient were sliced and the first as well as the last section of a series were stained with hematoxylin and eosin (H&E). For the processing of biopsies, a section from the middle of the tissue was stained with H&E. A lung pathologist determined the proportion of viable tumor cells, stromal cells, normal lung cells and necrotic areas. Total RNA was isolated using an AllPrep DNA/RNA/miRNA Universal Kit (Qiagen, Hilden, Germany) according to the manufacturer’s protocols. RNA quality was assessed by utilizing an Agilent 2100 Bioanalyzer and an Agilent RNA 6000 Nano Kit (Agilent Technologies, Boeblingen, Germany). With the Transcriptor First Strand cDNA Synthesis Kit (Roche, Basel, Switzerland) the total RNA was transcribed to complementary DNA and used for quantitative real-time polymerase chain reaction (qPCR). Complete description of the procedure is provided elsewhere [17].

### 4.3. Quantitative Real-Time PCR

5 µL of the cDNA (corresponding to 5 ng of isolated total RNA) was utilized for qPCR with the LightCycler480^®^ (Roche) in a 384-well plate format according to the Minimum Information for Publication of qPCR Experiments (MIQE)-guidelines [28]. Universal ProbeLibrary (UPL) assay (Roche) was used as amplification and detection system. Gene specific primers (TIB Molbiol, Berlin, Germany) were combined with the primaQuant 2 fold qPCR Probe-MasterMix (Steinbrenner Laborsysteme, Wiesenbach, Germany). Threshold cycle (C_t_)-values were evaluated with the LightCycler480^®^ software release 1.5 and the 2nd derivative maximum method (Roche). For the comparison of gene expression in tumor and non-malignant samples, relative expression of the genes (normalized to the houskeepers) was calculated (ΔCt values). The following primers were used for the detection of PAEP and the two housekeeper genes ESD and RPS18: PAEP forward (5′-cctgtttctctgcctacagga-3′), PAEP reverse (5′-cgtcctccaccaggactct-3′), ESD forward (5′-tcagtctgcttcag aacatgg-3′), ESD reverse (5′-cctttaatattgcagccacga-3′), RPS18 forward (5′-cttccacaggaggcctacac-3′), RPS18 reverse (5′-cgcaaaatatgctggaacttt-3′). The following UPL were used: PAEP (UPL77), ESD (UPL50), RPS18 (UPL46). Amplified qPCR products were cloned and sequenced to validate correct target amplification. The complete procedure is described elsewhere [17].

### 4.4. Immunohistochemistry

For the detection of glycodelin, the polyclonal N-20 antibody (sc-12289, Santa Cruz Biotechnology, Heidelberg, Germany) was used. Paraffin-embedded tissue sections were deparaffinized with the following steps: 2 × 10 min in xylol, 2 × 5 min in 100% ethanol, 1 × 3 min in 98% ethanol and 1 × 3 min in 70% ethanol. Antigen retrieval was performed in a steamer with sodium-citrate-buffer (10 mM sodium citrate, 0.05% Tween 20, pH 6.0) for 15 min. The staining procedure was performed with DAKO EnVision+ System-HRP (AEC) system (Dako, Hamburg, Germany). The detailed procedure and specificity controls of the glycodelin antibody are described elsewhere [17].

Staining was observed with an Olympus IX-71 inverted microscope. Pictures were taken with an Olympus Color View II digital camera and Olympus Cell-F software (cellSense dimension, V1.11, Olympus, Hamburg, Germany). Tiffs were assembled into figures using Photoshop CS6 (Adobe, San José, CA, USA).

### 4.5. Detection of Glycodelin in Human Sera

The glycodelin levels of the sera were detected using an enzyme-linked immunosorbent assay kit (ELISA BS-30-20, Bioserv Diagnostics, Rostock, Germany) with 50 µL of each serum in two technical replicates. In cases of tumor progression, glycodelin was measured in the sera collected during the patient’s routine checkup and/or before treatment. The readout and standard curve were performed with ELISA Reader (Tecan Group Ltd., Crailsheim, Germany). The results of the ELISA were visualized with GraphPad Prism 5 (GraphPad Software, San Diego, CA, USA).

### 4.6. Statistical Analyses

Data of qPCR and ELISA analyses were statistically analyzed under REMARK criteria [29] with SPSS 24.0 for Windows (IBM, Ehningen, Germany). The endpoint of the study was overall survival. Survival time was calculated from the date of diagnosis until the last date of contact or death. The cut-off for the glycodelin serum concentrations used for survival analyses was selected using the software tool “Cutoff-Finder” (http://molpath.charite.de/cutoff/index.jsp). Univariate analysis of survival data was performed according to Kaplan and Meier [30]. The log-rank test was used to test the significance between the groups. ELISA data were statistically analyzed with GraphPad Prism 5. The non-parametric, two-tailed Mann-Whitney *U* test [31] was used to investigate significant differences between the patient groups. Correlation of gene expression was performed using the nonparametric Spearman’s rank correlation analyses [32]. A *p*-value of less than 0.05 was considered significant.

## 5. Conclusions

In summary, we demonstrated that glycodelin is expressed in primary tumors as well as in metastases of NSCLC patients and that it is a promising biomarker for diagnosis, prognosis and especially for monitoring of disease. The measurement of glycodelin using an ELISA can be easily implemented in clinical setup and therefore optimize therapy adaption in case of a disease progression or recurrence.

## Figures and Tables

**Figure 1 cancers-10-00486-f001:**
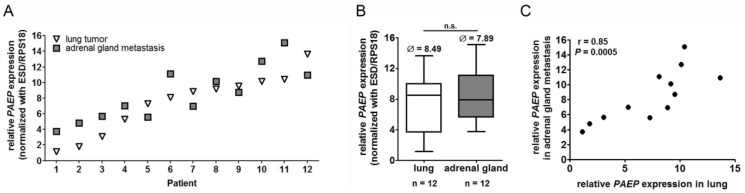
Glycodelin expression in primary tumors and adrenal gland metastases. 12 patient specimens (paired primary lung tumors and adrenal gland metastases) were investigated for *PAEP*/glycodelin expression. (**A**) Relative *PAEP* gene expression (ΔCt) in primary tumors and adrenal gland metastases. Please note that a lower value means a higher gene expression. (**B**) Median expression levels of *PAEP* in NSCLC tumors and adrenal gland metastases. (**C**) Nonparametric correlation analyses of *PAEP* gene expression. (**D**) Immunohistochemical tumor staining of patients from A) with a glycodelin specific antibody. A representative area is shown. *p* < 0.05 was considered as significant. n.s. = not significant, *PAEP* = progesterone-associated endometrial protein, NSCLC = non-small cell lung cancer.

**Figure 2 cancers-10-00486-f002:**
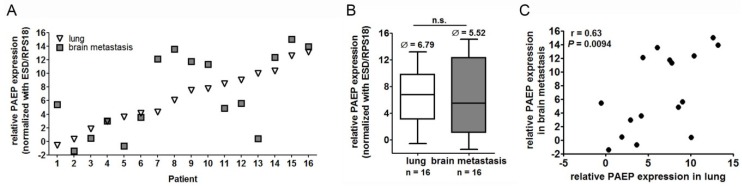
Glycodelin expression in primary tumors and brain metastases. 16 patient samples (primary lung tumors and brain metastases) were investigated for *PAEP*/glycodelin expression. (**A**) Relative *PAEP* gene expression (ΔCt) in primary tumors and brain metastases. Please note that a lower value means a higher gene expression. (**B**) Median expression levels of *PAEP* in NSCLC tumors and brain metastases. Ø: median (**C**) Nonparametric correlation analyses of *PAEP* gene expression. (**D**) Immunohistochemical tumor staining of 12 patients from (**A**) with a glycodelin specific antibody (FFPE tissue was not available for the other five patients). A representative area is shown. *p* < 0.05 was considered as significant. n.s. = not significant, *PAEP* = progesterone-associated endometrial protein, NSCLC = non-small cell lung cancer.

**Figure 3 cancers-10-00486-f003:**
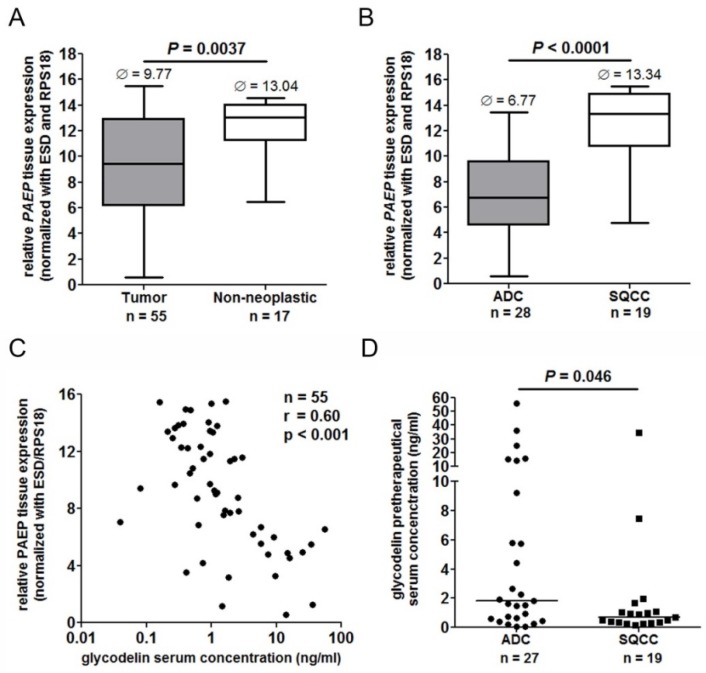
Glycodelin expression in biopsies and serum of advanced stage NSCLC patients. (**A**) Relative *PAEP* gene expression (ΔCt) in tumor biopsies ((*n* = 58) estimated tumor cell content ≥ 40%) compared to non-paired tumor-free lung biopsies (*n* = 17). Please note that a lower value means a higher gene expression. Ø: median (**B**) Relative *PAEP* gene expression in ADC and SQCC subhistologies of the cohort. (**C**) Correlation of *PAEP* expression in biopsies and measured glycodelin serum concentrations of the same patients (no serum available for 3 patients). *p* < 0.05 was considered as significant. (**D**) Glycodelin serum concentrations of ADC and SQCC patients (*n* = 46). *PAEP* = progesterone-associated endometrial protein, ADC = adenocarcinoma, SQCC = squamous cell carcinoma.

**Figure 4 cancers-10-00486-f004:**
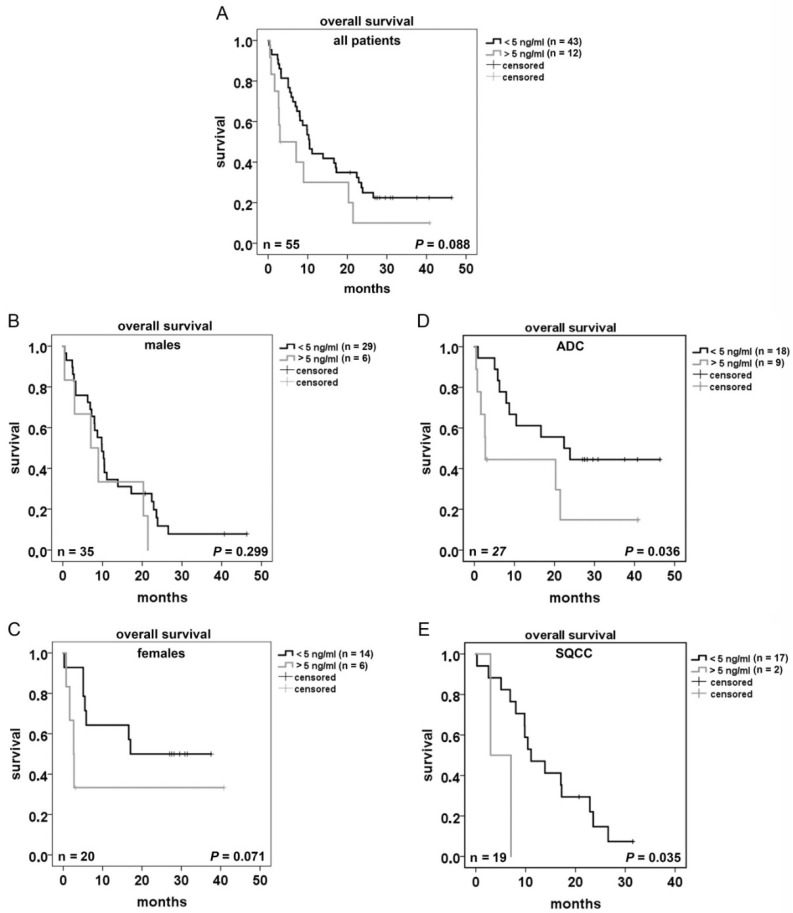
Prognostic potential of glycodelin serum concentrations in NSCLC advanced stage patients. (**A**–**E**) Kaplan-Meier curves using 5 ng/mL glycodelin serum concentration as the value to separate the patients in two groups. ADC = adenocarcinoma, SQCC = squamous cell carcinoma. *p* < 0.05 was considered as significant, *n* = number of patients.

**Figure 5 cancers-10-00486-f005:**
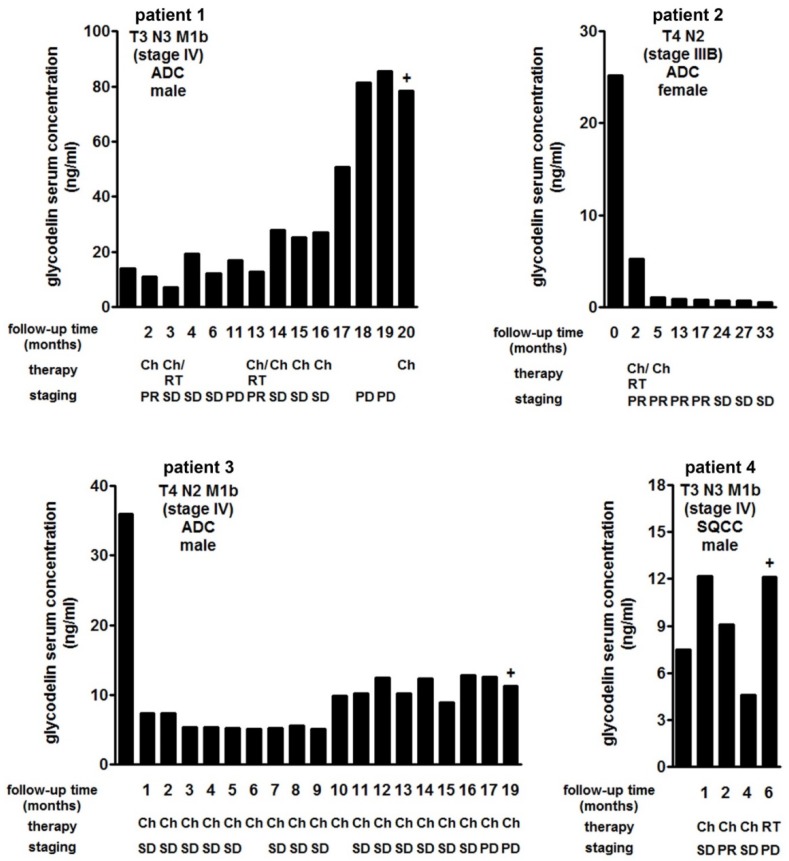
Glycodelin serum concentrations during patient follow-up. Glycodelin serum concentrations of four NSCLC patients during disease follow-up. Therapy response was defined by clinical staging. SD = stable disease, PR = partial remission, PD = progression disease, Ch = chemotherapy, RT = radiotherapy, + = time of death.

**Table 1 cancers-10-00486-t001:** Cohort characteristics.

Parameter	*n*	(%)	Parameter	*n*	(%)
Surgical Cohort	28		Biopsy Cohort (Malignant Only)	55	
Median age	61 (40–77)		Median age	66 (50–81)	
Gender	28	100	Gender	55	100
Male	18	64	Male	35	64
Female	10	36	Female	20	36
ECOG	28	100	ECOG	55	100
0	25	89	0	16	29
1	2	7	1	30	55
2	1	4	2	8	15
			n.d.	1	1
Histology	28	100	Histology	55	100
Adenocarcinoma	15	54	Adenocarcinoma	28	51
Squamous cell Carcinoma	3	11	Squamous cell Carcinoma	19	35
Large cell Carcinoma	5	18	Large cell Carcinoma	4	7
Other	5	18	Other	4	7
Initial clinical stage (7th edition)	28	100	Clinical stage (7th edition)	55	100
Stage IB	1	4	Stage IIIA	7	13
Stage IIA	6	21	Stage IIIB	11	20
Stage IIB	2	7	Stage IV	37	67
Stage IIIA	5	18			
Stage IIIB	3	11			
Stage IV	11	39			

n.d. not determined, ECOG: Eastern Cooperative Oncology Group Performance Status Scale.

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
