# Peer review of "Glycodelin as a Serum and Tissue Biomarker for Metastatic and Advanced NSCLC"

_cancers, 2018, doi:10.3390/cancers10120486_

Reviewer 1 Report

While the manuscript by Schneider et al. is interesting and important as it describes a potential new biomarker for NSCLC, there are some shortcoming that should be addressed.

Major comments:

1. The language needs to be revised. Especially there are multiple grammatical errors (including, but not restricted to, those in the use of past tense). Also mistakes like that in lines 51-52: were the patients metastasized or did the patients have metastasized cancer? As revision is needed througout the mansucript I do not list the corrections here.

2. As authors have themselves stated, some of the groups are extremely small for statistical comparison (i.e., in many cases contain samples from less than 10 patients and in one case only from two patients).

3. Glycodelin levels in serum vary greatly depending of the menstrual cycle stage: glycodelin expression in secretory stage endometrium is very high and this reflects the levels in circulation. Therefore, it would be important to know whether this has been taken into account. Looking the ages, it is likely that not all pateints are menopausal. In this context it would be intersting to know whether the ages of women in different groups of Figure 4C are different. What was the age (and menstrual cycle status) of patient 2 in Figure 5 when the first sample was obtained?

4. Glycodelin should be compared to currently used NSCLC markers, i.e., would glycodelin measurement be beneficial over these markers or give additional value if detected together with these markers?

5. Based on only four cases, showing very different profiles of glycodelin expression over time, the value of glycodelin as follow-up marker is questionable. You may mention that, but it would be good to tell that ”definite conclusion can not be drawn based on the these limited number of cases. However, this warrants further studies.”

6. Has the specificity of antibody staining be confirmed? There are no controls shown.

7. The primer sequences should be given, since without those the specificity (and which splice variants are detected) is not possible to evaluate.

Minor comments:

1. The authors state that a p-value<0.05 was considered significant. However, in abstract they say that ”…showed lower overall survival (P=0.088)”. If they want to include that it is better to say ”showed a tendency”  or use other similar expression.

2. In line refs 8,9: I would suggest to use a reference by some of the pioneers of glycodelin field (note that glycodelin has previously been most often referred as PP14). E.g., Seppälä M et al. Endocr Rev. 2002 Aug;23(4):401-30.

3. Lines 65-6: ”…was also seen to regulate...to be necessary…”, perhaps, rather than stating as a fact, you should state that ”has been suggested” since the study was done using a cell model.

4. In Table 1 the ”parameter names” (median age, gender etc.) do not have to be repeated for both cohorts.

5. For QRT-PCR results, I would suggest to use comparative Ct method (the 2^-ΔΔCt method) as the results expressed as ”fold-change” would be easier for the readers (Livak KJ, Schmittgen TD. Analysis of relative gene expression data using real-time quantitative PCR and the 2(-delta delta C(T)) method. Methods 2001;25:402–408).

6. Lines 166-167. The authors state that driver mutations had no influence on the serum levels of glycodelin. It is better to state that there were no association between these mutations and glycodelin levels.

7. It may be beneficial for the readers to describe various cancer relevant activities described for glycodelin and also tell more about the previous expression studies, especially those relating to lung cancer. For ovarian cancer Mandelin E et al., Cancer Res. 2003 Oct 1;63(19):6258-64 and for breast cancer Hautala LC et al.,  Breast Cancer Res Treat. 2011 Jul;128(1):85-95 would be better than the references included.

8. Should the conclusions be after the Discussion, not after the Matrials and Methods?

9. Line 260: was 5 ng or 5 µl of cDNA used? If 5ng, as the authors state, how this was measured?

10. Line 280, Should ELISA code be BS-30-20?

11. In references the Journal names should begin with capital letters (see also journal name in ref 11).

Author Response

1. The language needs to be revised. Especially there are multiple grammatical errors (including, but not restricted to, those in the use of past tense). Also mistakes like that in lines 51-52: were the patients metastasized or did the patients have metastasized cancer? As revision is needed througout the mansucript I do not list the corrections here.

The manuscript was revised by to improve the language quality.

2. As authors have themselves stated, some of the groups are extremely small for statistical comparison (i.e., in many cases contain samples from less than 10 patients and in one case only from two patients).

We agree that the cohort size is limited. We analyzed the glycodelin gene and protein expression in inoperable stage IIIB/IV patients using cryoconserved biopsies. Cryoconserved biopsies for research purposes are very rare and valuable. Our focus here was an exploration of glycodelin as a biomarker for late stage NSCLC with a combination of cryoconserved biopsies and matched blood samples to increase the value of our study. Future studies with more blood samples shall confirm the prognostic data of our current study.

3. Glycodelin levels in serum vary greatly depending of the menstrual cycle stage: glycodelin expression in secretory stage endometrium is very high and this reflects the levels in circulation. Therefore, it would be important to know whether this has been taken into account. Looking the ages, it is likely that not all pateints are menopausal. In this context it would be intersting to know whether the ages of women in different groups of Figure 4C are different. What was the age (and menstrual cycle status) of patient 2 in Figure 5 when the first sample was obtained?

You are right; the menstruation cycle might influence the serum concentration of glycodelin in female patients of the cohort. In fact, the female group of the cohort has a median age of 63 whereas the youngest female patient was 52 at time of biomaterial sampling. We cannot completely exclude, that any of the women might still have their menstruation cycle. We performed a correlation analysis (Pearson) of all measured values and the age of the female patients to address this. The result was r = 0.035. We therefore can be sure that there is absolutely no correlation between age and glycodelin serum concentration. We also addressed this point in the results and discussion part.

The age of the female patient in Figure 5 was 69.

4. Glycodelin should be compared to currently used NSCLC markers, i.e., would glycodelin measurement be beneficial over these markers or give additional value if detected together with these markers?

We will consider this suggestion in a future study. For the malignant pleural mesothelioma (MPM), we observed that a combination with another MPM biomarker (mesothelin) could increase the prognostic value. To our knowledge, there are no serum markers which are routinely used for the diagnosis of NSCLC. CYFRA 21-1 and CEA are described to be feasible markers and might be investigated in combination with glycodelin. 

5. Based on only four cases, showing very different profiles of glycodelin expression over time, the value of glycodelin as follow-up marker is questionable. You may mention that, but it would be good to tell that ”definite conclusion can not be drawn based on the these limited number of cases. However, this warrants further studies.”

We agree and addressed this point in the discussion.

6. Has the specificity of antibody staining be confirmed? There are no controls shown.

The specificity was shown in both publications from 2015 (Schneider et al., Clinical Cancer Research and Cancer Treatment Communications). We used the antibody blocking peptide to validate the specificity. Furthermore, we showed the siRNA mediated knockdown of glycodelin in Western blot analyses. We added the information in the material and method section.

7. The primer sequences should be given, since without those the specificity (and which splice variants are detected) is not possible to evaluate.

The primer and probe sequences are shown in the publications from 2015, but we added the information about the primers and probes in the material and method part. According to the MIQE-guidelines, we also performed a validation of the primer/probe combinations concerning PCR efficiency using 10-fold dilutions. In addition, we generally clone and sequence the amplified product of the primer/probe combinations to validate a correct amplification of the targets. The complete procedure is described in the Clinical Cancer Research publication from 2015.

Minor comments:

1. The authors state that a p-value<0.05 was considered significant. However, in abstract they say that ”…showed lower overall survival (P=0.088)”. If they want to include that it is better to say ” showed a tendency”  or use other similar expression.

We adressed this point.

2. In line refs 8,9: I would suggest to use a reference by some of the pioneers of glycodelin field (note that glycodelin has previously been most often referred as PP14). E.g., Seppälä M et al. Endocr Rev. 2002 Aug;23(4):401-30.

You are right. The group around Seppälä published a lot of work on the field of glycodelin. We revised the introduction part (see also point 7).

3. Lines 65-6: ”…was also seen to regulate...to be necessary…”, perhaps, rather than stating as a fact, you should state that ”has been suggested” since the study was done using a cell model.

We changed this part.

4. In Table 1 the ”parameter names” (median age, gender etc.) do not have to be repeated for both cohorts.

Since the parameter names partly differ between the columns, we prefer to keep them separately to have a uniform appearance of the table.

5. For QRT-PCR results, I would suggest to use comparative Ct method (the 2^-ΔΔCt method) as the results expressed as ”fold-change” would be easier for the readers (Livak KJ, Schmittgen TD. Analysis of relative gene expression data using real-time quantitative PCR and the 2(-delta delta C(T)) method. Methods 2001;25:402–408).

The ΔΔCt-method is used if a sample is compared to a control (to receive the fold-change, for example the gene expression change from a tumor and non-tumor sample). Here, we wanted to compare the relative expression of the gene in different groups. The shown data are therefore only the ΔCt-values (normalized to housekeepers). We clarified this in the material and method section.

6. Lines 166-167. The authors state that driver mutations had no influence on the serum levels of glycodelin. It is better to state that there were no association between these mutations and glycodelin levels.

We agree and changed the sentence.

7. It may be beneficial for the readers to describe various cancer relevant activities described for glycodelin and also tell more about the previous expression studies, especially those relating to lung cancer. For ovarian cancer Mandelin E et al., Cancer Res. 2003 Oct 1;63(19):6258-64 and for breast cancer Hautala LC et al.,  Breast Cancer Res Treat. 2011 Jul;128(1):85-95 would be better than the references included.

We revised the introduction part to add more information about the role of glycodelin in cancer.

8. Should the conclusions be after the Discussion, not after the Matrials and Methods?

The order of the sections is given by the journal.

9. Line 260: was 5 ng or 5 µl of cDNA used? If 5ng, as the authors state, how this was measured?

We used 5 µl cDNA which corresponds to 5 ng of isolated total RNA. We specified this sentence.

10. Line 280, Should ELISA code be BS-30-20?

You are right. We apologize for the mistake and corrected the product code.

11. In references the Journal names should begin with capital letters (see also journal name in ref 11).

We used the indicated style for the journal “Cancers” (MDPI style). Therefore, we can’t influence the spelling of the references.

Submission Date

05 November 2018

Date of this review

15 Nov 2018 13:10:45

Reviewer 2 Report

This manuscript is well written.

Minor point: In line33, PAEP expression was significantly upregulated in the tumor samples (P=0.0029). On the other hand, in Figure3A, p value is 0.0037. Which is correct?

Author Response

This manuscript is well written.

Minor point: In line33, PAEP expression was significantly upregulated in the tumor samples (P=0.0029). On the other hand, in Figure3A, p value is 0.0037. Which is correct?

You are right, we corrected the discrepancy.

Round  2

Reviewer 1 Report

Authors have revised the manuscript adequately and I do not have any further concerns.